# Grape Small Auxin Upregulated RNA (*SAUR*) 041 Is a Candidate Regulator of Berry Size in Grape

**DOI:** 10.3390/ijms222111818

**Published:** 2021-10-30

**Authors:** Ming Li, Rui Chen, Hong Gu, Dawei Cheng, Xizhi Guo, Caiyun Shi, Lan Li, Guoyi Xu, Shicao Gu, Zhiyong Wu, Jinyong Chen

**Affiliations:** 1Zhengzhou Fruit Research Institute, Chinese Academy of Agricultural Sciences, Zhengzhou 450009, China; liming07@caas.cn (M.L.); guhong@caas.cn (H.G.); chengdawei@caas.cn (D.C.); guoxizhi@caas.cn (X.G.); shicaiyun@caas.cn (C.S.); lilan01@caas.cn (L.L.); xuguoyi@caas.cn (G.X.); 82101182230@caas.cn (S.G.); 13598072703@163.com (Z.W.); 2Biotechnology Research Institute, Tianjin Academy of Agricultural Sciences, Tianjin 300192, China; chenrui.taas@gmail.com

**Keywords:** grape, berry size, cell expansion, *SAUR*

## Abstract

Grape (*Vitis vinifera*) is an important horticultural crop that can be used to make juice and wine. However, the small size of the berry limits its yield. Cultivating larger berry varieties can be an effective way to solve this problem. As the largest family of auxin early response genes, *SAUR* (small auxin upregulated RNA) plays an important role in the growth and development of plants. Berry size is one of the important factors that determine grape quality. However, the *SAUR* gene family’s function in berry size of grape has not been studied systematically. We identified 60 *SAUR* members in the grape genome and divided them into 12 subfamilies based on phylogenetic analysis. Subsequently, we conducted a comprehensive and systematic analysis on the *SAUR* gene family by analyzing distribution of key amino acid residues in the domain, structural features, conserved motifs, and protein interaction network, and combined with the heterologous expression in *Arabidopsis* and tomato. Finally, the member related to grape berry size in *SAUR* gene family were screened. This genome-wide study provides a systematic analysis of grape *SAUR* gene family, further understanding the potential functions of candidate genes, and provides a new idea for grape breeding.

## 1. Introduction

Grape (*Vitis vinifera*) is a perennial deciduous vine with abundant germplasm resources and widely distributed throughout the world. It can be used for fresh food, wine making, drying, and juice making [1]. The berry size is one of the important factors to determine the appearance quality of grape, and also the core factor that determines the merchantability of berry. In general, the size of grape under natural growth is difficult to meet the demand of consumers, so increasing berry size is of great significance for grape breeding [2]. At the same time, it can promote the healthy and sustainable development of the grape industry.

As the main target trait of long-term plant domestication, berry size plays an important role in improving the quality and yield of grape. As the demand for berries continues to grow, modern breeding took berry size as the main target trait. The berry size is determined by the number and volume of cells that make up the berry. The activity of cell division determines the final number of cells, but cell expansion is crucial. It can directly and effectively increase the berry volume and determine the final berry size [3]. In tomato, cell expansion contributes the most to the final berry size [4]. Two days after fertilization, the tomato ovules undergo rapid cell division to form a fleshy peel, which envelops the placenta and seeds to form young fruits. The transition from cell division to cell expansion is completed in 7–14 days after anthesis. Subsequently, the growth of the fruit mainly depends on cell expansion until the fruit matures, and the number of cells remains almost the same. Cell expansion occupies most stages of fruit development. The cell diameter of some tomato varieties can reach 0.5 mm, and diameter plays an important role in the final size of the fruit [5]. In *Arabidopsis*, the expansion of the valve determines the fruit size. The growth dynamics of *Arabidopsis* fruit were observed, and cell expansion and cell division data were collected for quantitative analysis. It was found that the valve cell had a higher rate of cell expansion, and no cell division was detected. Therefore, the fruit size of *Arabidopsis* mainly depends on cell expansion [6]. During the first rapid growth period of grape berry, cell division and cell expansion occurred simultaneously. At 4–6 weeks after anthesis, grapes entered the slow growth phase, the cell expansion rate decreased, and cell division stopped. Entering the second rapid growth period, the growth was completely dependent on cell expansion [7].

Auxin is an important hormone in plants, which plays a key role in plant growth and development. One of the functions of auxin is to stimulate cell expansion. It is this characteristic of auxin that allowed its discovery [8]. Previous studies found that auxin participates in many important processes by promoting cell expansion, including hypocotyl elongation, leaf growth, phototropic growth, apical hook, and hair development [9,10,11,12,13]. Under high temperature, PIF4 promoted hypocotyl elongation of *Arabidopsis* by regulating auxin synthesis and up regulate the expression of *SAUR* in hypocotyls [14,15]. Stern et al. increased the cell size of cherry fruit by more than 25% by applying 25 mg/L of 2, 4-d and 30 mg/L of NAA (1-Naphthaleneacetic acid) to the fruit [16]. The second stage of apple fruit development is the rapid cell expansion stage, in which the auxin content is significantly increased. Through the application of a low concentration of auxin, the cell size of apple fruit can be significantly increased [17]. The cell size of Japanese and loquat fruit treated with auxin increased significantly [18,19]. In the grape variety ‘Shiraz’, auxin can significantly increase berry size by promoting cell expansion [20]. VvCEB1 significantly increases the cell size of ‘Cabernet Sauvignon’ berry by regulating the expression of auxin-responsive genes [3].

Auxin plays a vital role in the process from embryo formation to senescence in plants, most of which is achieved by regulating auxin response genes. Early auxin response genes can be transcribed within a few minutes under the induction of auxin, and this induction does not require de novo protein synthesis [21]. Auxin early response genes can be divided into three categories: Aux/IAA (auxin/indole-3-acetic acid), GH3 (gretchen hagen3) and *SAUR* (small auxin up RNA) [22]. To date, significant progress was made in the functional research of Aux/IAA and GH3 proteins in the auxin response signaling pathway. Although the ability of *SAUR* to respond to auxin was known for a long time, little is known about their function and mechanism. *SAUR* was first found in elongating soybean hypocotyls induced by auxin [23]. *SAUR* was subsequently found in various plants such as *Arabidopsis*, tobacco, tomato, apple, litchi, and peach [24,25,26,27]. Some researchers found that *SAUR* participates in the shade avoidance response, high-temperature-induced growth and tropism caused by environmental stimuli such as light and gravity [14,28,29,30]. Further studies showed that *SAUR* participates in these processes by promoting cell expansion. Spartz et al. found that *SAUR019* could activate plasma membrane H^+^-ATP activity by promoting threonine phosphorylation at position 947 of the AHA2 protein, reduce the pH of the apoplast, and promote *Arabidopsis* cell expansion, resulting in elongation of the hypocotyl [31]. The results of radioisotope labeling and a taproot inhibition test showed that *SAUR063* increased auxin concentration in *Arabidopsis* hypocotyls by affecting auxin transport and promoted hypocotyl and stamen filament elongation [32]. Kong et al. found that *SAUR041* can regulate auxin transport activity, promote cell expansion, and then participate in the gravity response of roots [33]. Although the current research on *SAUR* function is mainly carried out in *Arabidopsis*, increasing evidence suggests that *SAUR* plays an important regulatory role in the process of cell expansion [34].

Cell expansion plays a key role in the process of ‘Hong yan Seedless’ berry expansion. During the two periods of rapid berry growth, the cells expanded obviously. In this study, we screened 60 *SAUR* genes based on conserved domain. Then, we conducted a comprehensive and systematic analysis on the *SAUR* family, including the distribution of key amino acid residues in the domain, phylogenetic relationships, structural features, conserved motifs, and protein interaction network, and combined with the sequencing results to screen candidate gene. Subsequently, we verified a potential molecular function of the gene. Our study provides insights that will be helpful for further study on function of grape genes in future and which may provide some candidate *SAUR* genes for predictable grape breeding.

## 2. Results

### 2.1. Identification, Annotation, and Chromosomal Location of the SAUR Family

The ORF of the *SAUR* family obtained from the Phytozome database was analyzed, and the genes without the *SAUR* conserved domain were removed. Subsequently, we used SMART (http://smart.embl-heidelberg.de/ accessed on 29 September 2021) and DNAStar software (Version 7.1, DNASTAR Company, Madison, WI, USA) to analyze the amino acid and nucleotide sequence, leaving only the member with the longest sequence in the similar sequence. Finally, we selected 60 members for further study according to the sequencing results. We collated the details of the *SAUR* family from Phytozome (Appendix A). The *SAUR* family members were mapped on 19 chromosomes according to their position on the chromosome and were named from *VvSAUR001* to *VvSAUR060* (Figure 1). We further analyzed the function of the *SAUR* family, especially the members related to berry expansion. The results showed that the *SAUR* family members were distributed on only 12 chromosomes. We found that the number of members on different chromosomes varied greatly. For example, chromosome 3 contains 29 members, while chromosomes 1, 2, 12, 16, and 19 contain only one member each.

### 2.2. Phylogenetic Relationships of the SAUR Family

The SAUR proteins from grape and *Arabidopsis* were selected for the construction of an unrooted phylogenetic tree based on the multiple sequence alignments of the conserved domains (Figure 2) to better understand not only the evolutionary relationship between VvSAUR and AtSAUR proteins but also the function of the grape *SAUR* family. We made appropriate adjustments based on Heim’s method. For example, the VII subfamily was divided into VII a, VII b, VII c, and VII d. Finally, the grape *SAUR* family was divided into 12 subfamilies according to the 79 *SAUR* members of *Arabidopsis* [35]. It turned out that there is no *SAUR* member in the subfamilies VII and VIII. This might be the result of long-term evolution. The results showed that the number of family members in different subfamilies varied greatly. For example, the largest subfamily VII b contained 20 members, while the smallest subfamilies V, VI, and VII d contained only one member each. The classification of the grape *SAUR* family provided evidence of their evolutionary relationship.

### 2.3. Detection of the Conserved Motif and Gene Structure

The motif structure determined the classification of family members and played a key role in the interaction of different modules in transcriptional complexes and transduction of signals. We analyzed the distribution of 10 conserved motifs in 12 *SAUR* subfamilies and counted their sequences, length, and number of occurrences of motifs (Appendix A and Appendix A). The motif types ranged from 3 to 7 and the average number of motifs ranged from 2 to 5 for 12 subfamilies. Each type of motif appeared only once in each gene. Some motifs are universal, 57 members have motif 2, and 48 members have motif 1. Some motifs are conservative and only appear in a particular subfamily. For example, motif 10 exists only in subfamily IV (*VvSAUR048* and *VvSAUR051*). There is no motif in subfamily II (*VvSAUR060*) and subfamily VI (*VvSAUR058*). This may be why each subfamily has its own specific function.

Introns can appear anywhere in a gene and are important for exon replication and ultimately affect gene function. The phase-0-intron is located between the third nucleotide of the leading codon and the first nucleotide of the following codon. The phase-1-intron is located between the first nucleotide and the second nucleotide of a codon. The phase-2- intron is located between the second nucleotide and the third nucleotide of a codon. A Symmetric exon is an exon that occurs between two identical phases. If it is an integral multiple of three, there is no code shift mutation, so it can be copied; otherwise, it cannot be copied. The symmetric exons and phase-0-intron can facilitate exon shuffling, recombinational fusion and protein domain exchange [36]. The results showed two phase-1-symmetric exons and four phase-2-symmetric exons among 106 exons, but no phase-0-symmetric exon, and 48 introns included 10 phase-0-introns, 22 phase-1-introns and 15 phase-2-introns (Appendix A). The analysis of the conserved motif and gene structure explains the diversity of the *SAUR* family and provides a further theoretical basis for the study of its functions.

### 2.4. The Conserved Amino Acid Residues in the SAUR Domain

We performed multiple sequence alignment and counted the amino acid ratio of the core sequence of the *SAUR* family domain for learning its function (Figure 3). The results showed that the amino acid residues in the conserved core region of the *SAUR* domain were consistent with a previous report [37]. Among them, eight conserved amino acid residues had a consensus rate higher than 75%.

### 2.5. The Synteny Analysis of the SAUR Family

We constructed a synteny relationship graph to analyze the origin of the *SAUR* family and its evolutionary relationship in grape, *Arabidopsis* and tomato (Figure 4). The *SAUR* family members synteny relationship between grape and *Arabidopsis* were one to one, one to multiple. and multiple to one, respectively. Therefore, 21 *VvSAUR* genes and 29 *AtSAUR* genes constitute 41 pairs of *SAUR* genes (Appendix A). It indicated that large-scale expansion occurred prior to the divergence of *Arabidopsis* and grape, and some *VvSAUR* genes share a common ancestor with *AtSAUR* gene counterparts. According to a previous study, the Solanaceae ancestors experienced an independent whole-genome triplication (WGT) event after differentiation, while grape did not experience any additional genome-wide replication event [38]. The results of synteny analyses showed that the number of tomato *SAUR* family members was less than three times that of grape. This indicated that a large number of gene losses occurred in the whole-genome duplication (WGD) of tomato. We found 7 and 40 paralogous *SAUR* gene pairs in grape and tomato according to a collinear correlation of the *SAUR* gene, but not in tomato (Appendix A). It is speculated that some redundant genes with similar functions may be lost during evolution.

### 2.6. Expression Characterization of the VvSAUR Family at Different Developmental Stages

The berries from five periods were used for sequencing and analysis. Among the 60 members of the *SAUR* family, 47 *SAUR* genes distributed in 11 subfamilies are expressed at least one of the developmental stages of berry (Figure 5). The FPKM values were used to estimate the expression characterization of the *SAUR* family for screening the candidate gene associated with berry expansion. The results showed that eight genes (*VvSAUR006*, *VvSAUR008*, *VvSAUR016*, *VvSAUR035*, *VvSAUR039*, *VvSAUR047*, *VvSAUR049,* and *VvSAUR060*) were mainly expressed in the first rapid growth period, and seven genes (*VvSAUR022*, *VvSAUR040*, *VvSAUR042*, *VvSAUR051*, *VvSAUR052*, *VvSAUR053* and *VvSAUR056*) were mainly expressed in the second rapid growth period. Three genes (*VvSAUR003*, *VvSAUR041* and *VvSAUR046*) were mainly expressed in both rapid growth stages, but the expression level in the first rapid growth stage was higher than that in the second rapid growth stage. *VvSAUR041* belongs to the VII B subfamily and may be associated with berry expansion according to previous studies.

### 2.7. Function Prediction of Candidate Genes

We were only interested in genes expressed in both rapid growth phases, so we analyzed the interaction network of *VvSAUR003*, *VvSAUR041,* and *VvSAUR046* using STRING based on *VvSAUR* orthologs in *Arabidopsis* (Figure 6). It could help us understand gene function and efficiency [39]. *VvSAUR003* (*AT1G19840* in *Arabidopsis*) is a SAUR-like auxin responsive protein. Its interaction protein YAB5 belongs to the YABBY family and is responsible for regulating the initiation of embryonic shoot apical meristem (SAM) development. *VvSAUR041* (*AT4G38840* in *Arabidopsis*) is involved in the response to auxin stimulus. *SAUR63* belongs to the ARG7 family and may promote auxin-stimulated organ expansion, such as hypocotyls, stamen filaments and petals. *HBI1* is a typical *bHLH* transcription factor that acts as a positive regulator of cell expansion downstream of multiple external and endogenous signals by direct binding to the promoters and activation of the two expansion genes *EXPA1* and *EXPA8*, encoding cell wall loosening enzymes. Transcriptional activity is inhibited when binding to the *bHLH* transcription factor *IBH1*. *VvSAUR046* (*AT2G36210* in *Arabidopsis*) is an auxin-responsive SAUR protein. Its interacting protein is SAG21. It can mediate tolerance to oxidative stresses by minimizing the negative effects of oxidation and monitoring photosynthesis during stress, promote root development and prevent premature aging. In addition, it is involved in the resistance against compatible pathogens such as *Botrytis cinerea* and *Pseudomonas syringae* pv. tomato. These analyses suggest that *VvSAUR041* may be involved in berry expansion.

### 2.8. Subcellular Localization

To determine the subcellular localization of *VvSAUR041*, the GFP protein was fused to the C terminus of *VvSAUR041* with a mutational stop codon under the control of the CaMV35S constitutive promoter. The *35S:: VvSAUR041:GFP* fusion protein and control *35S:: GFP* were transferred into the *Arabidopsis* protoplast using a PEG-mediated procedure. The control *35S:: GFP* was distributed throughout the whole cell, whereas the *35S:: VvSAUR041:GFP* fusion protein was detected in the cytoplasm of the *Arabidopsis* protoplast (Figure 7).

### 2.9. VvSAUR041 Promotes Seedling Growth and Cell Expansion of Arabidopsis

The T3-generation homozygous transgenic *Arabidopsis* was used to further analyze the biological function. The main inflorescence stem height of the three OE lines (OE-3, OE-6 and OE-7) was higher than that of VC lines at stage 6 [40] (Figure 8B). We selected 10 plants from each line to calculate the rosette diameter in the lower panel, and the results showed that three OE lines had a larger rosette compared to that of VC lines (119%, 124% and 119%, respectively) (Figure 8C,D). Then, we measured the cell area of the third leaf section of 3-week-old *Arabidopsis* by electron microscope. Compared with VC lines, the average cell area was noticeably greater in the three OE lines (120%, 136%, and 118%, respectively) (Figure 8E,F). In addition, for each line, we selected 10 siliques to calculate the length and found that the siliques of the three OE lines were longer than those of VC lines (117%, 128%, and 129%, respectively) (Figure 8G,H). These results revealed that *VvSAUR041* plays an essential role in promoting seedling growth and cell expansion.

### 2.10. VvSAUR041 Plays a Positive Role in the Fruit Development of Tomato

To study the role of *VvSAUR041* in the process of fruit development, 10 fruits from each line were selected for statistics on their fruit weight. The result showed that the weight of the three OE lines (OE-2, OE-5, and OE-6) was greater than that of the VC lines (121%, 129% and 131%, respectively) (Figure 9C). In addition, we selected 10 fruits from each line to evaluate the fruit volume by measuring the displacement of the fruits after they were placed in a measuring cylinder. The fruit volume of the three OE lines was greater than that of the VC lines (118%, 137%, and 138%, respectively) (Figure 9D). Compared with the VC lines, the average cell area was greater in the pericarp tissue of the three OE lines (126%, 131%, and 138%, respectively) (Figure 9E,F). These results suggested that *VvSAUR041* plays a positive role in the fruit development of tomato.

## 3. Discussion

As one of the most important horticultural crops in the world, grape has important nutritive value in addition to its use in fresh food, winemaking, and juice making. Berry size is a key factor in the commercialization of grapes, as well as an important selection indicator for breeding. Cell size plays a decisive role in the final berry size. Auxin has the function of promoting cell expansion. Rayle et al. proposed the acid growth theory to explain this phenomenon [41]. However, the acid growth theory lacks strong genetic evidence. An auxin early response gene, *SAUR*, can be transcriptionally induced by auxin within minutes without the need for de novo synthesis of the protein. The research on the biological functions of *SAUR* mainly focuses on hypocotyl growth, stem elongation, and auxin transport [42]. Spartz et al. found that *SAUR019* activates plasma membrane H^+^-ATPases by inhibiting PP2C-D phosphatases to promote cell expansion, which provides evidence for the acid growth theory. These results suggest that *SAUR* plays a key role in cell expansion.

The PinotNoir PN40024 genome was used as a query to blast the peptide with the BLASTp algorithm. All genes with non-redundant hits and expected values less than 0.001 were considered for further analysis. We obtained 63 *VvSAUR* genes, removed the pseudogenes through the analysis of the conserved domain and finally screened 60 *VvSAUR* genes. We named them according to their positions on chromosomes. The results of the chromosome distribution showed that there were three *SAUR* gene clusters on chromosomes 3 and 4, suggesting that tandem gene duplication probably played a pivotal role in *SAUR* gene expansion in the grape genome (Figure 1). *SAUR* gene clusters have also been reported in rice, maize, and tomato, which indicates that species-specific expansion is common in the *SAUR* family [43]. It was reported that there is rapid gene expansion in gene families related to morphological development and stress response, and gene expansion is usually accompanied with plant evolution under environmental factors [44]. Then, we conducted a phylogenetic analysis of *VvSAUR* and divided it into 12 subfamilies, the results of which are similar to those for *Arabidopsis* and tomato [45]. The results showed that *VvSAUR041* belongs to subfamily VII b (Figure 2). *AtSAUR068* (*AT5G18010*) of subfamily VII a regulates cell expansion by activating PM H^+^-ATPases to facilitate apoplast acidification and mechanical wall loosening, resulting in tomato hypocotyl elongation [46]. *AtSAUR008* (*AT1G29440*) of subfamily VII d promotes auxin-stimulated organ elongation. The result of motif and gene structure provides strong evidence for the phylogenetic relationship of the *VvSAUR* family (Appendix A). The conservation of the *VvSAUR* family domain shows that they still retain their basic functions during the long evolutionary process (Figure 3).

Comparative genomics is an effective tool for rapidly understanding, locating, and cloning unknown genes. There are genes with similar location and sequence on the homologous chromosomes of interspecies and intraspecies, and the synteny provides a theoretical basis for them [47]. A large number of collinear regions in the genome provide strong evidence for the existence of whole-genome duplication (WGD) [48]. Ohno et al. first explained how a diploid genome evolved into a tetraploid genome through polyploidy hypothesis, and then, Wolfe et al. confirmed this hypothesis in *Saccharomyces cerevisiae* genome [49]. The genome of *Arabidopsis* underwent two WGDs (WGD: α and β) and one WGT (WGT: γ) during its evolution [50]. Subsequently, the grape genome was studied, and its ancestral genome may be paleohexaploid [51]. These results laid a foundation for the study of grape molecular biology. This is helpful to study the origin and evolution of grape and the localization and cloning of important functional genes. Among auxin early response genes, the *SAUR* family has the largest number of members, which is essential for normal plant growth and adaptation to the external environment. However, the results of comparative genomics showed that members of the grape *SAUR* family were lost during evolution (Figure 4). It is speculated that the efficiency of the auxin response pathway may be improved by the loss of some functional redundant genes. It is now more than 30 years since *SAUR* was discovered. Due to the redundancy of its functions, the functions of most of its members are unknown. To further understand the role of *SAUR* in fruit development, we analyzed the expression levels of *SAUR* during five critical stages of fruit development. The results showed that the expression levels of *VvSAUR003*, *VvSAUR041,* and *VvSAUR046* increased during the two rapid growth phases (Figure 5). Then we used *Arabidopsis* orthologs to predict their interaction regulation network (Figure 6). The predicted interaction genes *SAUR63*, *HBI1,* and *IBH1* of *VvSAUR041* are involved in organ expansion. In the remaining two gene interaction regulatory networks, no interaction genes were predicted to be directly related to organ expansion. According to the above results, it is speculated that *VvSAUR041* plays a positive role in the process of berry expansion.

The localization of proteins is closely related to their functions. Previous studies showed that *SAUR* may be located in a variety of structures such as the nucleus, cytoplasm, mitochondria, chloroplast, and plasma membrane [52,53]. In this study, *VvSAUR041* was located in the cytoplasm, and it is speculated that it can regulate the plant life activities. To further study the regulation mechanism of *VvSAUR041* in berry development, we used genetic transformation technology to study its function. The statistical results showed that the vegetative organ and silique of the transgenic *Arabidopsis* line were larger than those of the VC line. Then, we measured the cell area of the leaves, and the results were consistent with the phenotype. We compared the final size of tomato fruit and found that the volume of the transgenic line was larger than that of the VC line, which is consistent with the above analysis. In addition, the weight of transgenic lines was greater than that of VC lines. Observation of tomato pericarp tissue cells showed that the cell area of the transgenic line was larger than that of the VC line. These results indicate that larger cells eventually lead to larger fruits in transgenic tomato. This study provides a theoretical basis for further understanding the function of the *SAUR* family members in the development process of grape berry.

## 4. Materials and Methods

### 4.1. Sequence Retrieval and Feature Analysis

The protein sequences of *Arabidopsis* and tomato were obtained from Phytozome (https://phytozome.jgi.doe.gov/pz/portal.html, accessed on 5 March 2020). We identified the grape SAUR protein with the HMMER software version 3.0 based on Hidden Markov Model (HMM) profiles of the auxin-inducible signature domain structure (PF02519) and Pfam database (Pfam: http://pfam.sanger.ac.uk/, accessed on 9 March 2020). The reference genome PinotNoir PN40024 (http://plants.ensembl.org/Multi/Tools/Blast?db=core, accessed on 16 March 2020) was used to blast the peptide with the BLASTp algorithm. The *SAUR* genes with nonredundant hits and expected values less than 0.001 were reserved for analysis. We used a conserved domain database (https://www.ncbi.nlm.nih.gov/Structure/cdd/wrpsb.cgi, accessed on 18 March 2020) to further analyze the conserved domain and finally obtained 60 *SAUR* genes. The Phytozome and ExPASy Proteomics server (http://web.expasy.org/compute_pi/, accessed on 19 March 2020) were used to obtain details of the *SAUR* family. SMART (http://smart.embl-heidelberg.de/, accessed on 21 March 2020) and DNAStar were used to analyze nucleotide and amino acid sequences.

### 4.2. The Sample Collection and RNA-Seq

The grape (Vitis vinifera. L. ‘Hong yan Seedless’) cultivar was grown in the experimental field of the Zhengzhou Fruit Research Institute, Chinese Academy of Agricultural Sciences, Zhengzhou, Henan Province, China (34°43′ N, 113°39′ E, altitude 111 m). The berries at 16 (S1), 20 (S2), 23 (S3), 48 (S4), and 79 (S5) days after full bloom (DAFB) were harvested for RNA-Seq (S2 and S4 were the first rapid growth stage and the second rapid growth stage, respectively). A group of 30 berries was replicated three times. Total RNA was extracted using the TIANGEN RNAprep Pure Plant Kit (Tiangen Biotech, Beijing, China), and 1% agarose gel was used to detect RNA degradation and contamination. A NanoPhotometer^®^ spectrophotometer was used to detect RNA purity (IMPLEN, CA, USA). A Qubit^®^ RNA Assay Kit in Qubit^®^ 2.0 Flurometer was used to measure the RNA concentration (Life Technologies, California, USA). A Nano 6000 Assay Kit of the Bioanalyzer 2100 system was used to assess RNA integrity (Agilent Technologies, California, USA). Illumina Hiseq™ obtained raw data, which were then transformed into the original sequence through CASAVA base recognition analysis. After the low-quality sequences were removed, the data were compared with the reference genome through HISA T2 (https://daehwankimlab.github.io/hisat2/, accessed on 25 March 2020), and the results were compared by RSeQC (http://rseqc.sourceforge.net/, accessed on 28 March 2020). The fragments per kilobase per million (FPKM) value was used for expression analysis.

### 4.3. Bioinformatic Analysis

We used MapDraw V2.1 to locate the *SAUR* family on chromosomes of grape based on the Phytozome database. We carried out phylogenetic analysis by using Clustal Omega (http://www.ebi.ac.uk/Tools/msa/clustalo/, accessed on 29 March 2020) and the phylogenetic tree was displayed using iTOL (https://itol.embl.de/itol.cgi, accessed on 29 March 2020). The numbers are bootstrap values based on 1000 iterations. Only bootstrap values larger than 50% support were displayed. The multiple sequence alignment of *VvSAUR* was performed by CLUSTALW with default parameters. The intron/exon structure of the *SAUR* family members was analyzed using GSDS2.0 (http://gsds.cbi.pku.edu.cn/, accessed on 2 April 2020). The conserved motifs were identified by MEME (http://meme-suite.org/index.html accessed on 2 April 2020), and e-values < 1 × 10^−20^ were retained for further analysis. We used OrthoMcl (https://orthomcl.org/orthomcl/, accessed on 22 March 2020) to display the relationships of orthologous and paralogous *SAUR* genes among grapes, *Arabidopsis* and tomato. P-value cut-off of 1 × 10^−5^ was chosen for all clusters constructed. The heatmap of the *SAUR* family was displayed with TBtools (https://github.com/CJ-Chen/TBtools/releases, accessed on 26 March 2020). The target protein sequence was uploaded to the online server STRING to predict the interaction network (https://string-db.org/cgi/input.pl, accessed on 28 March 2020) with option value > 0.700.

### 4.4. Subcellular Localization

The open reading frame (ORF) with mutational stop codon was cloned between the Xba I and Sal I sites of the pB221-GFP vector with the T4 DNA ligase (Thermo Scientific, Waltham, MA, USA). The recombinant and control plasmids were transformed into *Arabidopsis* leaf protoplasts as described previously [54]. After 18 h, the GFP fluorescence was observed under a laser scanning confocal microscope (Olympus FV1000 viewer, Tokyo, Japan), 488 nm, argon-ion laser excitation, 507 nm detection GFP. Chloroplast autofluorescence was analyzed using 488 nm argon-ion laser excitation, SP 630 nm IR detection, a pinhole of about 1.0 unit and an optical section thickness of about 0.5 µm.

### 4.5. The Transformation of Arabidopsis and Tomato

We removed genomic DNA from total RNA by DNase I according to the manufacturer’s instructions (Thermo Scientific, Waltham, MA, USA) and first-strand complementary DNA (cDNA) synthesis was performed with the RevertAid First Strand cDNA Synthesis Kit (Thermo Scientific, Waltham, MA, USA). We obtained 294 bp full-length ORFs of *VvSAUR041*. The products were purified and integrated into the blunt vector (pEASY-Blunt Simple Cloning Kit, Beijing, China) for sequencing. The ORF was cloned between the Nco I and Bgl II sites of the pCAMBIA3301 vector with the T4 DNA ligase. We used the floral dip method to transfer the recombinant plasmid and control plasmid into *Arabidopsis* Columbia ecotype [55]. The RT-qPCR and phosphinothricin resistance were used to screen positive plants. Homozygous T3-generation plants were used for functional verification. The ORF was cloned between the Xho I and EcoR I sites of the pART-CAM-EGFP vector with T4 DNA ligase. We obtained positive micro-tom plants by *Agrobacterium tumefaciens*-mediated transformation [56]. The kanamycin and timentin resistance was used to screen positive plants. The plants infected with control plasmid were taken as the control. The tomato plants were placed into identical pots containing mixed soil (soil:vermiculite:peat = 3:1:1) in a greenhouse according to the environmental conditions described previously [57]. Homozygous T2-generation plants were used for further study.

### 4.6. Morphological and Cytological Observations

The fruit volume and weight were measured at 54 days post anthesis (dpa) according to the methods described previously [58]. At the mature green stage (45 dpa), the fruit is almost the same size as that at maturity and the pericarp cell is more rigid, with a less variable reading [59]. To cytologically assess the pericarp cell of the fruit, morphological and cytological observations were performed as previously described with some modifications using samples taken from tomato fruit at 45 dpa. The leaf of *Arabidopsis* and fresh fruit of tomato were soaked in FAA solution for 24 h. The samples were dehydrated under an ethanol gradient. The wax-soaked tissue was embedded in an embedding machine (JB-P5, Wuhan, China). The wax blocks were sliced into 4 μm with a paraffin slicer (RM2016, Shanghai, China). Subsequently, the sections were dewaxed with xylene and stained with toluidine blue, and neutral gum sealing was performed. Finally, the images were observed and captured using a microscope (OLYMPUS-DP71, Tokyo, Japan). In each line, nine section views were used to calculate the cell area with three repetitions. We measured and calculated the results using Image Pro plus 6.

### 4.7. Analysis of RT-qPCR

The total RNA extraction and the preparation of the first-strand cDNA were consistent with the previous description. RT-qPCR was performed in the presence of SYBR green qPCR Master Mix (Thermo Scientific, Waltham, MA, USA) and the amplification was performed in a Real-Time PCR System (LightCycler^®^480 system, Basel, Switzerland). All reactions were repeated three times. The primers were designed using Oligo 7.0 (Appendix A). The *SlActin* gene was used as an internal control for grape [60]. *AtUBQ3* was used as an internal control for *Arabidopsis* [61].

### 4.8. Statistical Analysis

All experiments were replicated independently at least three times, and data are shown as the mean ± SD of three independent experiments. Data were subjected to a statistical analysis according to the Student’s t-test and significant differences among the means were determined by the least significant difference (LSD) method, * at *p* < 0.05 and ** at *p* < 0.01.

### 4.9. Accession Numbers

Sequence data from this work can be found in the NCBI database (SRA data: PRJNA749809).

## 5. Conclusions

Using ‘Pinot Noir’ genome as the reference genome, we performed high-throughput sequencing on five critical stages of berry development in ‘Hong yan Seedless’. Then, we analyzed the members of the *SAUR* family based on sequencing results and bioinformatics analysis. The results let us to speculate that *VvSAUR041* may be related to cell expansion. Finally, we used *Arabidopsis* and tomato genetic transformation technology to study its function and found that *VvSAUR041* can promote cell expansion. In this way, we can fully explore the original data and combine bioinformatics analysis with molecular biology experiments more efficiently for studying the function of *SAUR* family in berry development.

## Figures and Tables

**Figure 1 ijms-22-11818-f001:**
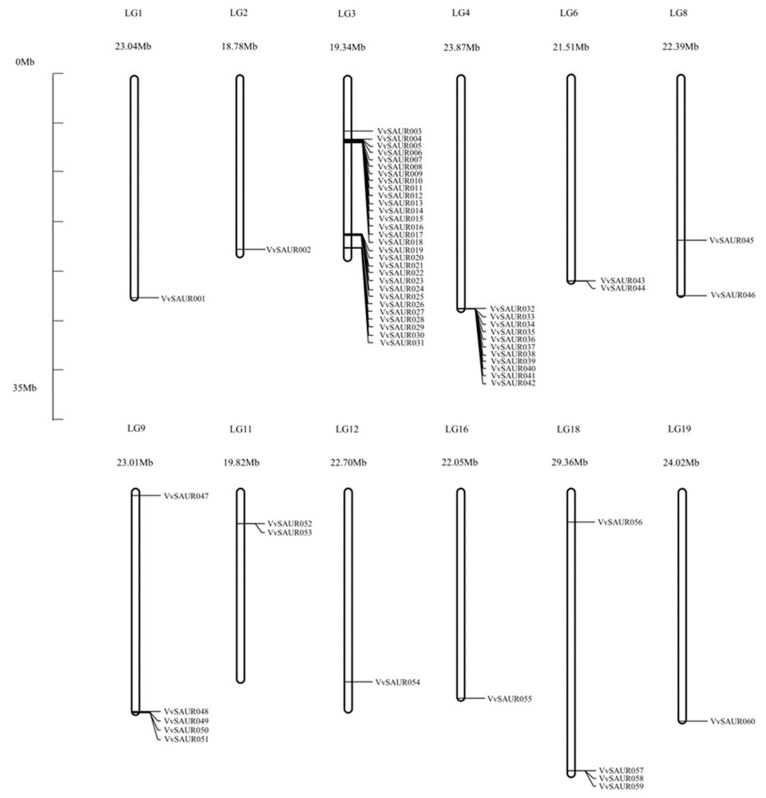
Distribution of 60 *VvSAUR* genes on grape chromosomes. Centromeric positions are shown according to location of each *VvSAUR* gene.

**Figure 2 ijms-22-11818-f002:**
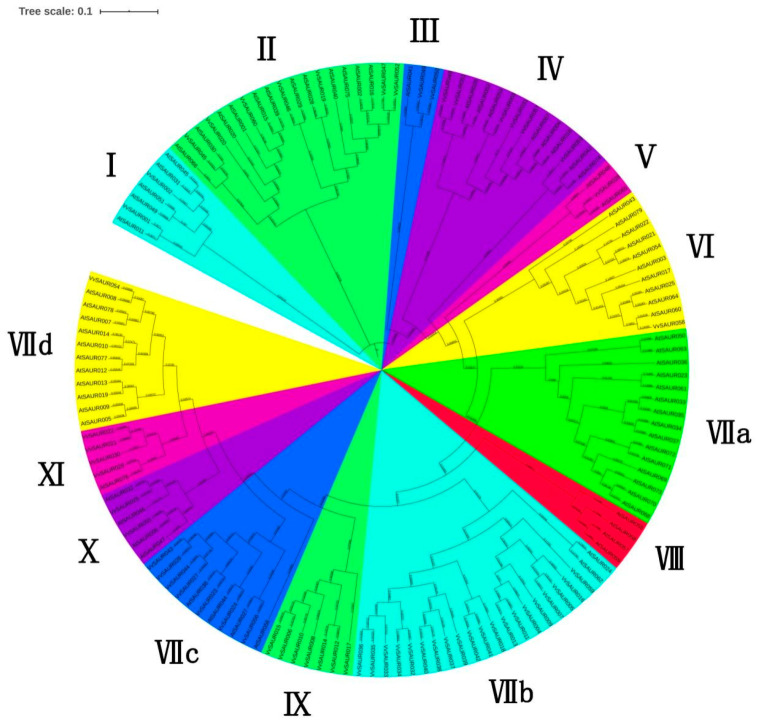
An unrooted phylogenetic tree of the *VvSAUR* family. Phylogenetic tree was constructed online with Interactive Tree of Life (iTOL). Numbers are bootstrap values based on 1000 iterations. Only bootstrap values larger than 50% support are displayed.

**Figure 3 ijms-22-11818-f003:**
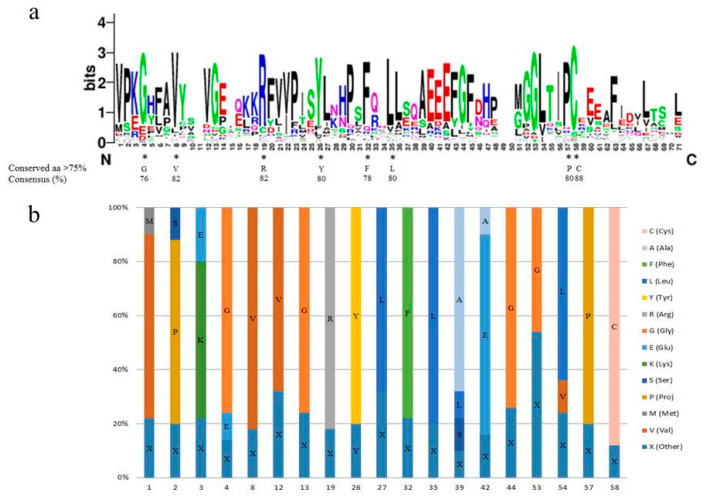
Characterization and distribution of *VvSAUR* domain. (**a**) Sequence logo of *VvSAUR* domain. Asterisks represent conserved amino acids that contain more than 75%. (**b**) Distribution of amino acids in the *VvSAUR* domain. Columns of different colors represent percentage of amino acids at this site.

**Figure 4 ijms-22-11818-f004:**
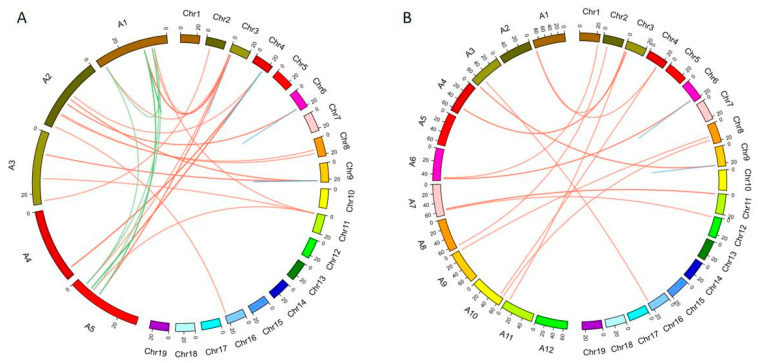
Analysis of paralogous *SAUR* genes and their orthologues in grape, *Arabidopsis* and tomato. (**A**) Analysis of paralogous and orthologues in grape (Chr1–Chr19) and *Arabidopsis* (A1–A5). Blue lines represent paralogous *SAUR* genes in grape. Green lines represent paralogous *SAUR* genes in *Arabidopsis*. Red lines represent their orthologous *SAUR* genes. (**B**) Analysis of paralogous and orthologues in grape (Chr1–Chr19) and tomato (A1–A12). Blue lines represent paralogous *SAUR* genes in grape. Red lines represent their orthologous *SAUR* genes. Relationships of orthologous and paralogous *SAUR* genes among grape, *Arabidopsis* and tomato were drawn by Circos 0.69.6 (http://circos.ca/, access on 23 May 2020).

**Figure 5 ijms-22-11818-f005:**
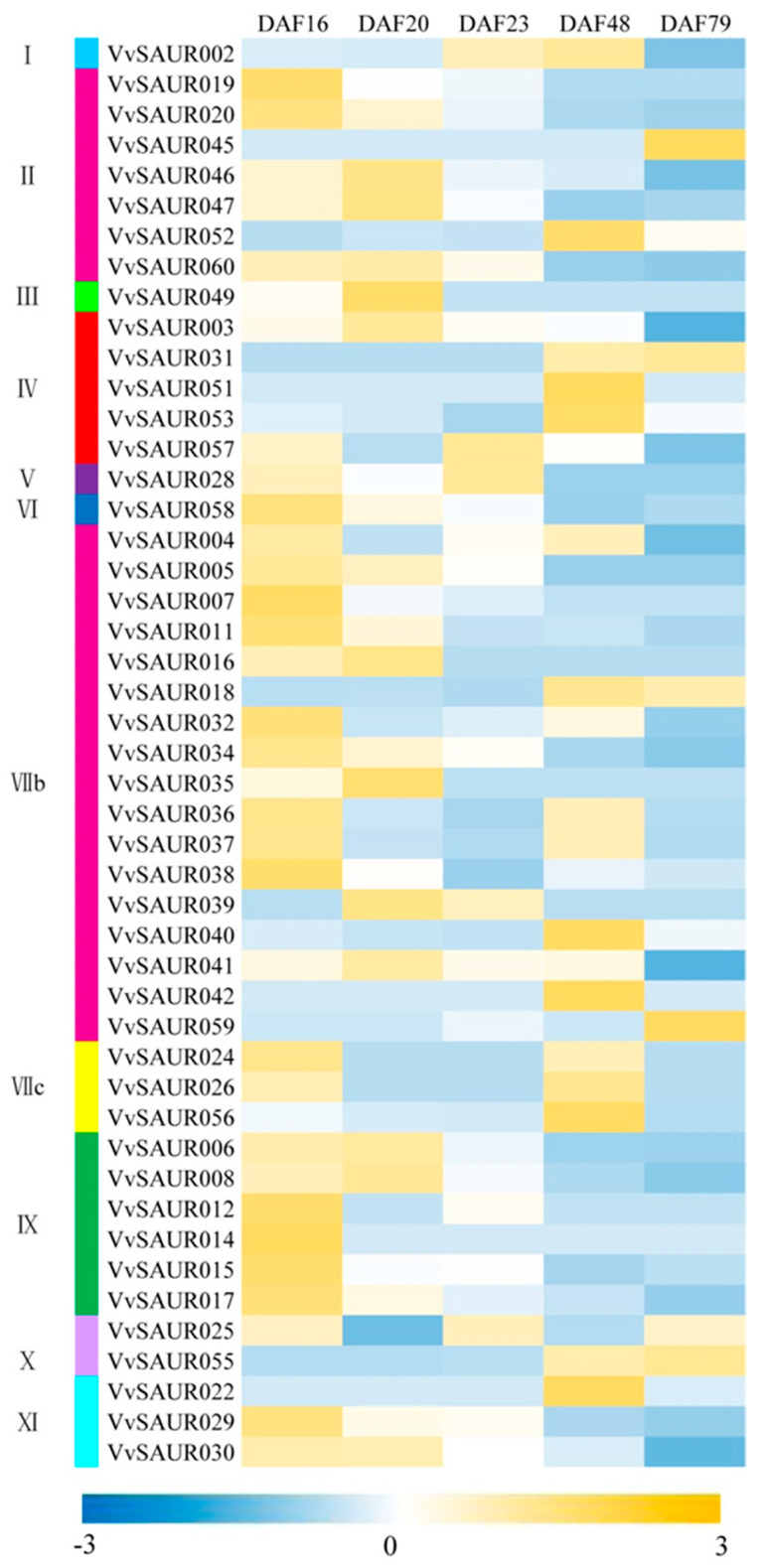
*VvSAUR* family heatmap of ‘Hong yan Seedless’ berry for five critical development periods.

**Figure 6 ijms-22-11818-f006:**
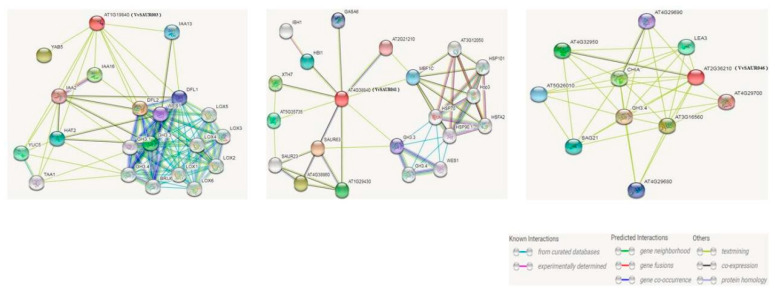
Interaction network analysis for *VvSAUR003*, *VvSAUR041* and *VvSAUR046*. Note: The predicted results are based on the orthologous gene in *Arabidopsis*. *VvSAUR* genes are shown in parentheses.

**Figure 7 ijms-22-11818-f007:**
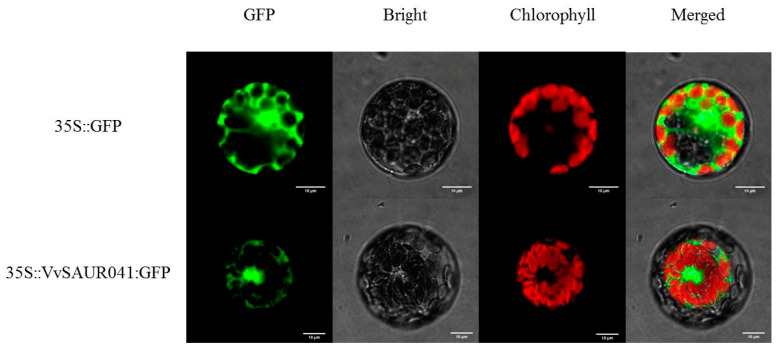
Subcellular localization of *VvSAUR041.* Vector control (*35S::GFP*) and fusion protein construct *35S:: VvSAUR041:GFP* were introduced into Arabidopsis protoplast.

**Figure 8 ijms-22-11818-f008:**
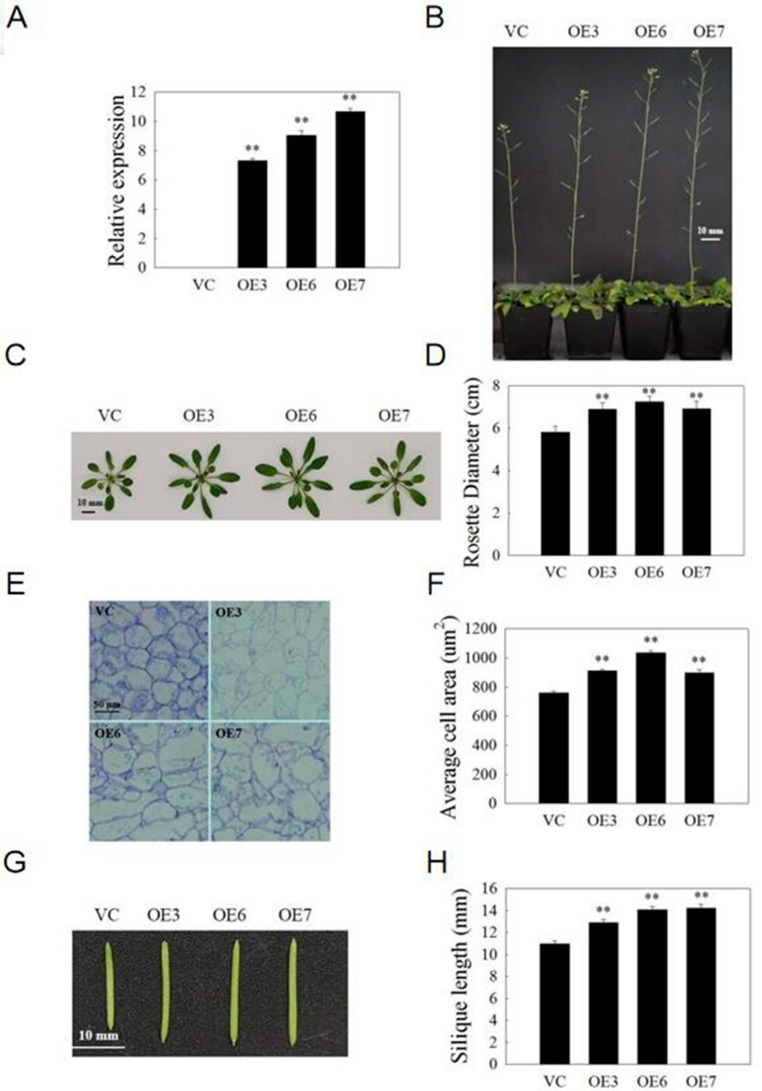
*VvSAUR041* promotes seedling growth and cell expansion of *Arabidopsis*. (**A**) Transcription level of *VvSAUR041* in OE and VC transgenic *Arabidopsis* based on RT-qPCR. (**B**) Phenotype of main inflorescence stem at stage 6. (**C**) Rosette leaf phenotype of 4-week-old *Arabidopsis*. (**D**) Rosette diameter of *Arabidopsis* in lower panel. (**E**) Third leaf section of 3-week-old *Arabidopsis*. (**F**) Average cell area of 3-week-old *Arabidopsis*. (**G**) Phenotype of silique. (**H**) Length of silique. Asterisks indicate a significant difference (** *p* < 0.01).

**Figure 9 ijms-22-11818-f009:**
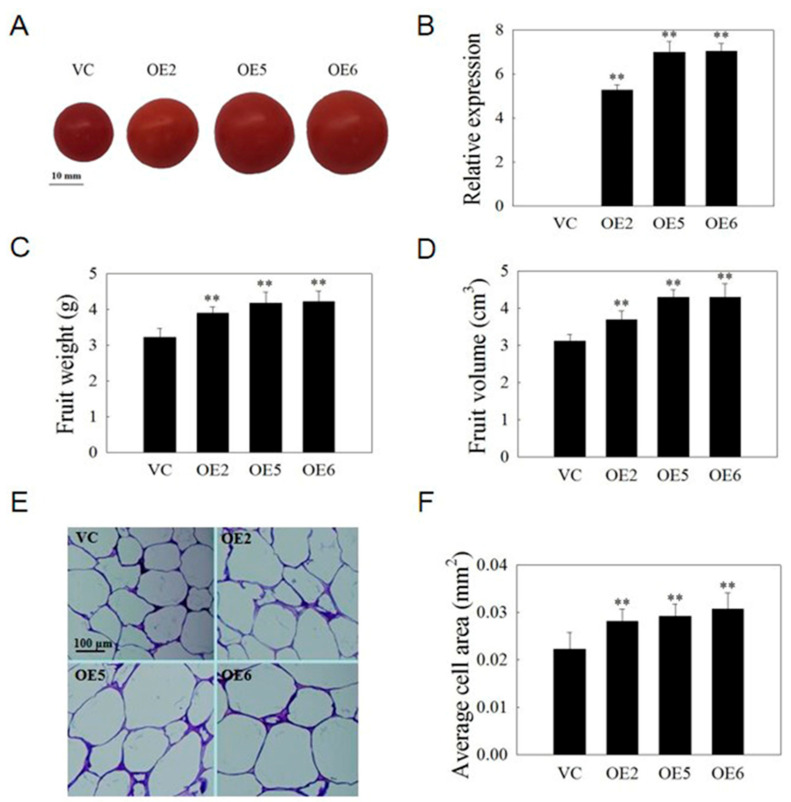
*VvSAUR041* plays a positive role in fruit development of tomato. (**A**) Fruit of mature tomato. (**B**) Transcription level of *VvSAUR041* in OE and VC transgenic tomato based on RT-qPCR. (**C**) Fruit weight of mature tomato. (**D**) Fruit volume of mature tomato. (**E**) Image of pericarp tissue at mature green stage. (**F**) Average cell area of tomato pericarp. Asterisks indicate a significant difference (** *p* < 0.01).

## Data Availability

Sequence data from this work can be found in the NCBI database (SRA data: PRJNA749809).

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
