# Peer review of "Grape Small Auxin Upregulated RNA (SAUR) 041 Is a Candidate Regulator of Berry Size in Grape"

_ijms, 2021, doi:10.3390/ijms222111818_

Round 1
Reviewer 1 Report
The manuscript titled “VvSAUR041, a Small Auxin-up RNA-Like Gene, Regulates Berry Size in Grape” identified the auxin early responsive SAUR family in grapes, and investigated the chromosomal locations, motif conservation, phylogenetic relationships, and expression under different developmental stages of these SAURs. The authors were able to identify one significant candidate from these SAURs, SAUR041, the overexpression of which appeared to increase cell area and silique size in Arabidopsis and fruit size in tomato. Please see comments below.
Selected major comments are listed below:
- SAUR genes from Arabidopsis and tomato are not annotated? Why the authors need to identify “SAUR protein with the HMMER software” from these species?
- More method details are needed:
- More details on how RNA-seq was performed and analyzed are needed, e.g. sequencing equipment version, number of reads, mapping percentage, any statistical test, etc.
- More details on tomato transformation are needed.
- Please provide corresponding grapefruit pictures when the RNA-seq samples were harvested.
- To claim that VvSAUR041 is localized to the nucleus, the authors will need to show the position of the nucleus, e.g. by staining the nucleus.
- The authors used Arabidopsis orthologs of their grape gene candidates to build interaction network. Have the authors checked if all the interacting partners are present in grape genome?
- For both Arabidopsis and tomato, the overexpression of VvSAUR041 increased the fruit size, do these overexpression lines show any other difference to the VC, e.g. slower growth? What might be compensated for the gain in fruit size?
- Please provide figures with high resolutions, and revise the figure captions to include the abbreviations, statistical significance, replicate/data point information.
Specific comments:
- Line 108: Please specify what database was used.
- Line 116-119: Any thought why SAURs are preferentially distributed on Chr LG3?
- Line 143-153: Background information for what the 10 conserved motifs are is needed.
- Line 173-175: How did the author conclude the two leucine residues contribute to protein folding and function? Have the authors check the structure of any SAUR proteins if available?
- Line 175-177: What is the reasoning for this statement? Did the author check if the amino acid change at N- or C-terminal affect signal peptide?
- Line 184-200: This entire section is very confusing. Please revise accordingly to make the point straightforward.
- Fig 8B: Please show the entire plants for all the lines.
- Line 495: The authors have shown data for VvSAUR041 only in Arabidopsis and tomato, better not use language like “VvSAUR041 can promote berry development through cell expansion”, please revise the title accordingly as well.
- Does the Methods go before the Conclusions or after?
Reviewer 2 Report
Li et al. set out to investigate the role of small auxin upregulated (SAUR) genes in grape fruit development, reasoning that auxin is often involved in controlling organ size in plants, and that SAURs could participate in this auxin signaling pathway. They begin with extensive genomic analyses of SAUR paralogues in the grape genome, identifying, naming, and classifying 60 SAUR genes. Based on their analyses, they decided to pursue whether one specific SAUR gene, SAUR041, might promote growth. They then show that heterologously overexpressing SAUR041 in Arabidopsis and tomato is sufficient to stimulate growth at several developmental stages in these species.
Overall, this work shows some promise, but I think that there are several major concerns that need to be addressed.
- I am not clear why SAUR041 was selected over the other SAUR I’ve read the manuscript a few times, and I still can’t understand the reasoning. This needs to be better explained, OR the authors need to say that they picked this one at random among a few options (the other candidates being SAUR003 and SAUR046, which are also expressed during rapid growth phases).
- The manuscript would have more clear impact if the authors tested the effect of overexpressing SAUR046 and SAUR003 on growth, too. This would reveal whether SAUR041 is, indeed, a special gene involved in auxin responses, or if many SAUR genes have similar impacts.
- Most importantly, the authors never directly test their major hypothesis that SAUR041 is involved in grape berry development. The title, abstract, and much of the text is very misleading. They show that SAUR041 is expressed in berries, and that overexpressing SAUR041 in other species is sufficient to stimulate some additional growth, but they never demonstrate whether SAUR041 expression causes or even contributes to berry growth in grapes. Therefore, the manuscript requires significant revision to focus on what the authors actually tested, and not on what they speculate.
Relatively minor comments:
Introduction:
Auxin promotes cell division and expansion, depending on context. It is not “most typical” that it promotes cell expansion; auxin plays key roles in cell division at meristems.
The summary of auxin signaling is very haphazard and incomplete; I do not know why ABP1, the atypical and controversial auxin receptor, is mentioned instead of the canonical TIR family of auxin receptors; nor why PIF4 regulation of auxin biosynthesis is discussed instead of any of the myriad other studies of regulators of auxin biosynthesis and signaling. This section should be more focused on what is relevant for this study, and otherwise general and cite the most important work on auxin—not a random selection of studies.
It isn’t immediately obvious to me that studies of tomato or Arabidopsis fruit development have strong relation to grape fruit development; could the authors better describe the relationship among these species and their fruits?
It also isn’t immediately obvious to me why the authors chose to focus on SAUR genes instead of any number of other auxin signaling genes. Is there a particular reason to specifically focus on these?
Figures 1, 2, and 4 all address similar topics, and might be better if presented as separate panels of a single figure to improve the likelihood that readers are interested (it feels, as is, like no one of these figures has much information, but together they might make more of an impact).
Figure 1, the labels are nearly impossible to read.
Figure 2, I recommend making panel a into two lines, because it is very hard to read as it is. Panel B is challenging because the colors are sometimes too similar (E, S, X, and L are all slightly different shades of blue). Again, the labels are an incredibly small font and nearly impossible to read.
Figure 5: I find the heatmap red and green colors hard to see. I recommend picking another color palette to be more accessible to a broad range of readers. Also, I think that black here signifies intermediate expression, but black usually means no expression; so I think having white as the intermediate would be better, or maybe using a color scale like Viridis.
Figure 6 is probably the least interesting of the figures. These computational predictions based on orthology to Arabidopsis don’t really tell us much about what is happening in grape, and certainly don’t contribute to the larger story about SAURs and grape development—they just suggest that SAURs are related to auxin signaling. I recommend removing this figure to improve focus.
Figure 7: Panel A: What is the goal of showing this gel? I’m not sure what the different numbers mean, are these just from bacteria? That’s not really useful information. Panel B: You do not need to show the sequence of GFP here. Panel C: I am not convinced that a GFP-fused SAUR localizes to the same place as the endogenous protein or reflects endogenous protein activity. For example, Spartz et al. (2012, Plant Journal) showed that SAUR protein stability is dramatically increased by N-terminal GFP tagging; I’m not familiar enough with this field to know what other efforts have been made in the past to discover functional features of SAUR proteins. Moreover, the protoplast localization shown doesn’t make much sense; we see strong fluorescence in an internal body surrounded by chloroplasts, but not at the cell periphery. This could be nuclear localization, vacuolar localization, or some other unintended effect in protoplasts, but it is certainly not simply cytosolic & nuclear localization. I recommend including counterstains (such as a nuclear stain, like propidium iodide, Hoechst stains, or DAPI) to show where the nucleus is in these cells, and showing several independent replicates so that the readers know this is not an artifact.
Figure 8: Panel A: If expression is “0” in the mock/vector control, then what does the y-scale represent? Was SAUR041 expression itself tested, or was some tag tested instead? Panel F: What cells were measured? How many individual plants were examined? General comments: Overall, I agree that it seems that overexpressing this SAUR promotes larger plant growth.
Figure 9: As above, panel B suggests that there was zero expression in vector control, but then says “5” as relative expression—relative to what? Something has to be “1”. Again, as above, I think these results overall support the hypothesis that SAUR041 overexpression promotes growth.
Lastly, as a small note, it should be "RT-qPCR", not "qRT-PCR". The PCR step is quantitative, not the RT.
Round 2
Reviewer 2 Report
Thank you for your thorough consideration of my suggested revisions to improve this manuscript. I appreciate many of the changes that you made, and I am glad to see that we agree on most points. Although I might disagree about some of the data presentation (e.g., how Figures 1, 2, and 4 are organized), in the end, that is your decision to make.
I have only two major concerns remaining.
- The title is still extremely misleading and inaccurate. We do not know from your study that VvSAUR041 regulates berry size in grape. We do learn from your experiments that ectopically expressing VvSAUR041 in heterologous systems, tomato and Arabidopsis, can stimulate cell expansion. An acceptable alternative might be, "Grape SMALL AUXIN UPREGULATED RNA (SAUR) 041 is a candidate regulator of berry size in grape", or "Ectopic expression of VvSAUR041, a small auxin upregulated RNA gene in grape, can promote fruit expansion in heterologous models".
To put this another way, we already know that auxin can stimulate cell expansion in diverse organisms, and that SAURs can contribute to this. We do not know that SAUR041 is responsible for cell expansion in grape from this study. Mutants lacking SAUR041 would be needed to confidently demonstrate this. Even overexpressing SAUR041 in grape would not be sufficient evidence for the original title.
Please also change the manuscript throughout to be more modest. For two examples:
In the abstract, there is a sentence that says this gene was "screened and verified"; verified to do what, and how?
l. 100: "verified its function by heterologous expression in Arabidopsis and tomato"; this verifies a potential molecular function of the gene, but does not verify that the gene is involved in this process in grapes.
2. I still object to the phrase "most typical" on line 55. As the authors agree in their response, the precise impact of auxin depends on context, and cell expansion is one--but only one--impact of auxin. In other developmental and physiological contexts, auxin has no impact on cell expansion, but strongly impacts cell division.
Author Response
Reviewer 2
Thank you for your letter and for reviewers' comments concerning our manuscript entitled "VvSAUR041, a small auxin-up RNA-like gene, regulates berry size in grape."(Manuscript ID: ijms-1401779). Those comments are all valuable and very helpful for revising and improving our paper, as well as the important guiding significance to our researches. We have studied comments carefully and have made correction which we hope meet with approval. The main revise to comments are as flowing:
Q The title is still extremely misleading and inaccurate. We do not know from your study that VvSAUR041 regulates berry size in grape. We do learn from your experiments that ectopically expressing VvSAUR041 in heterologous systems, tomato and Arabidopsis, can stimulate cell expansion. An acceptable alternative might be, "Grape SMALL AUXIN UPREGULATED RNA (SAUR) 041 is a candidate regulator of berry size in grape", or "Ectopic expression of VvSAUR041, a small auxin upregulated RNA gene in grape, can promote fruit expansion in heterologous models".
To put this another way, we already know that auxin can stimulate cell expansion in diverse organisms, and that SAURs can contribute to this. We do not know that SAUR041 is responsible for cell expansion in grape from this study. Mutants lacking SAUR041 would be needed to confidently demonstrate this. Even overexpressing SAUR041 in grape would not be sufficient evidence for the original title.
Thank you again for your valuable suggestions. It has been modified as required.
Q 2 Please also change the manuscript throughout to be more modest. For two examples:
In the abstract, there is a sentence that says this gene was "screened and verified"; verified to do what, and how?
- 100: "verified its function by heterologous expression in Arabidopsis and tomato"; this verifies a potential molecular function of the gene, but does not verify that the gene is involved in this process in grapes.
- I still object to the phrase "most typical" on line 55. As the authors agree in their response, the precise impact of auxin depends on context, and cell expansion is one--but only one--impact of auxin. In other developmental and physiological contexts, auxin has no impact on cell expansion, but strongly impacts cell division.
Thank you again for your valuable suggestions. It has been modified as required.
In new version, red indicates changes. If you have any queries, please don’t hesitate to contact me as soon as possible.
Thank you for your valuable opinions
Thank you and best regards.
Yours sincerely,
Ming Li
E-mail: liming07@caas.com
Tel: 18736056917